# A warm-up strategy with or without voluntary contraction on athletic performance, lower-leg temperature, and blood lactate concentration

**Seunghee Lee[1], Junhyeong Lim[2], Jihong Park[3]***

**1** Division of Sports Science & Medicine, Graduate School of Physical Education, Athletic Training Laboratory, Kyung Hee University, Yongin, Korea, **2** Physical Education, Graduate School, Athletic Training Laboratory, Kyung Hee University, Yongin, Korea, **3** Department of Sports Medicine, Athletic Training Laboratory, Kyung Hee University, Yongin, Korea

* Jihong.park@khu.ac.kr

**Data Availability Statement:** All relevant data are within the paper and its supporting information files.

## Abstract

It is unclear whether temperature-related warm-up effects can be accomplished by passive warm-up (e.g., by external heat). Therefore, this study compared the effects of two different warm-up protocols with and without voluntary contraction on subsequent sprinting and jumping performance. Eighteen healthy male collegiate students (23.3 ± 2.4 years, 173.8 ± 7.2 cm, 70.5 ± 9.3 kg) randomly experienced 10 min of active (jogging on a treadmill; belt speed: 9.0 km/h at a 1% incline) and passive warm-up (lying down in the warm-up chamber; inner ambient temperature set at 35°C) protocols, followed by ten sets of intermittent exercises in two separate sessions. Athletic performance, lower-leg muscle temperature, and blood lactate concentration were statistically compared using analysis of variance with Tukey-Kramer post-hoc comparisons. Cohen's d effect sizes (ES) were also calculated. There was no warm-up protocol effect over time on 20 m sprint times (condition × time: $F_{9,323} = 1.26$, $p = 0.25$). Maximal vertical jump heights were different (condition × time: $F_{9,323} = 2.0$, $p = 0.04$) such that subjects who performed the active warm-up protocol jumped higher (51.4 cm) than those who did the passive warm-up (49.2 cm, $p = 0.04$). There was a warm-up protocol effect over time on lower-leg muscle temperature (condition × time: $F_{12,425} = 13.99$, $p < 0.0001$) in that there was a 5.5% and 5.8% increase after active (32.8 to 34.6°C, ES = 2.91) and passive (32.9 to 34.9°C, ES = 3.28) warm-up protocols, respectively. Blood lactate concentration was different (condition × time: $F_{2,85} = 3.61$, $p = 0.03$) since the values at the post-warm-up measurements were different between warm-up conditions (active: 4.1 mmol/L; passive: 1.5 mmol/L, $p = 0.004$, ES = 1.69). Subsequent sprint and jump performance did not differ between the duration- and muscle temperature-matched active and passive warm-up protocols. Non-thermal effects from the warm-up activity may be minimal for sprinting and jumping performance in recreationally active males.

**Funding:** The author(s) received no specific funding for this work.

**Competing interests:** The author(s) have declared that no competing interests exist.

## Introduction

A warm-up is a well-recommended practice performed prior to any physically demanding activity [1, 2]. As given by its name, the warm-up has been defined as "organisms promote work more effectively at higher temperatures" [3]. The performance improvement due to warm-up activities is largely attributed to temperature-related mechanisms such as increased core [4] and muscle [5] temperature. Specifically, a 2% (0.7˚C) core temperature increase improved sprinting and jumping performance by 6% and 4%, respectively [4]; an increase in working muscle temperature of just 1˚C could affect the variation of athletic performance up to 5%. In general, muscle temperature increases rapidly during the first 3 to 5 min of warm-up and reaches a plateau after 10 to 20 min of activity [6]. Warm-up protocols with voluntary muscle contractions such as running have been defined as active warm-up [7]. Alternatively, the use of external heat (e.g., heat garments) also permits elevation of body temperature [8, 9]. This type of warm-up protocol has been defined as passive warm-up [7].

The effects and mechanisms of both warm-ups on subsequent athletic performance could be explained by thermal and non-thermal factors. First, temperature rise is caused by metabolic heat production [10] and energy release [11, 12] due to muscle contraction in active warm-up, while heat gain occurs through convection (e.g., air) or conduction (e.g., thermal device) in passive warm-up; thus, the relative contributions of the body's regulatory systems in active and passive warm-up methods are different: more cardiac [13], respiratory [14], and circulatory [15] change occur during an active warm-up, resulting in more energy expenditure [9]. This conversely indicates that the capacity of high-intensity exercise may be preserved for later tasks by performing a passive warm-up. Second, non-thermal effects are hardly expected in the passive warm-up. Independent of the increase in body temperature, these can be achieved by preconditioning (active contraction) of the muscle [16]. Specifically, increased spinal-reflex excitability [16–18], decreased muscle stiffness [16], postactivation potentiation (PAP) [11, 19], and improved psychological preparedness [16] are known effects from the literature. Considering these non-temperature related benefits, well-prepared explosive [16] or sport-specific [11, 20] activities after passive warm-up is not likely expected.

According to the pros and cons of both types of warm-ups with or without voluntary muscle contraction associated with the thermal and non-thermal effects discussed above, one would prefer to perform an active warm-up due to its non-thermal effect. However, the relative contribution of the non-thermal effect to the improvement in athletic performance does not seem to be significant [21]. To elicit a non-thermal effect such as postactivation potentiation (PAP), a high-intensity warm-up activity is required [7]. This process is likely accompanied by fatigue, which requires a rest duration for subsequent performance [7]. Therefore, if the magnitude of thermal-effect and subsequent athletic performance following warm-up protocols with or without voluntary contraction is similar, either warm-up condition would be optionally acceptable. If the difference in the non-thermal effect following either warm-up type is not large, the next question to consider regarding which type of warm-up leads to optimal athletic performance would be this: Should the thermal effect be achieved by voluntary contraction? A direct comparison on athletic performance, such as a series of sprints and jumps following temperature-rise matched warm-up conditions with or without voluntary contraction, allows athletes and coaches to determine specific warm-up protocols prior to a particular situation or sport.

Hence, the current study compared the effects of a 10-min active warm-up (jogging on a treadmill) to a time and thermal effect–matched passive warm-up (lying in a warm-up chamber) on subsequent 20 m sprint and countermovement jump performance, lower-leg temperature, and blood lactate concentration during intermittent high-intensity exercises. Secondarily, fatigue perception, heart rate, and energy expenditure were recorded. We

hypothesised that athletic performance and lower-leg temperature during intermittent exercises following either warm-up protocol would not be different. We also expected that the levels of blood lactate concentration, fatigue perception, heart rate, and energy expenditure following the active warm-up would be higher than those of the passive warm-up.

## Materials and methods

### Study design and experimental approach

Subjects randomly experienced two different warm-up protocols (active—jogging on the treadmill, and passive—lying prone in the warm-up chamber) during two separate sessions, 48 h apart (Figs 1 and 2). The order of warm-up protocols was determined by tossing a coin during the pre-warm-up measurements at the first session. The effects of each warm-up protocol were examined by analysing subsequent performance changes in 20 m sprints and maximal vertical jumps during high-intensity intermittent exercises. To obtain a similar thermal effect (e.g., temperature increase) on both warm-up protocols, the inside ambient temperature of the warm-up chamber was set to 35°C. This setup was based on the results of a preliminary pilot study (n = 5) on monitoring temperature rises of two warm-up protocols. To examine the thermal effects of both warm-up protocols, lower-leg skin and muscle temperatures were recorded at pre- and post-warm-up measurements.

The average air temperature and relative humidity in the laboratory (where the treadmill and chamber were in) and the lower-extremity track (where the intermittent exercises were performed) were recorded as 25.5 ± 1.1°C and 43.5 ± 6.8%, respectively.

### Subjects

Eighteen healthy, recreationally active male collegiate students (age: 23.3 ± 2.4 years, height: 173.8 ± 7.2 cm, mass: 70.5 ± 9.3 kg) who were exercising on a regular basis (> 150-min/week)

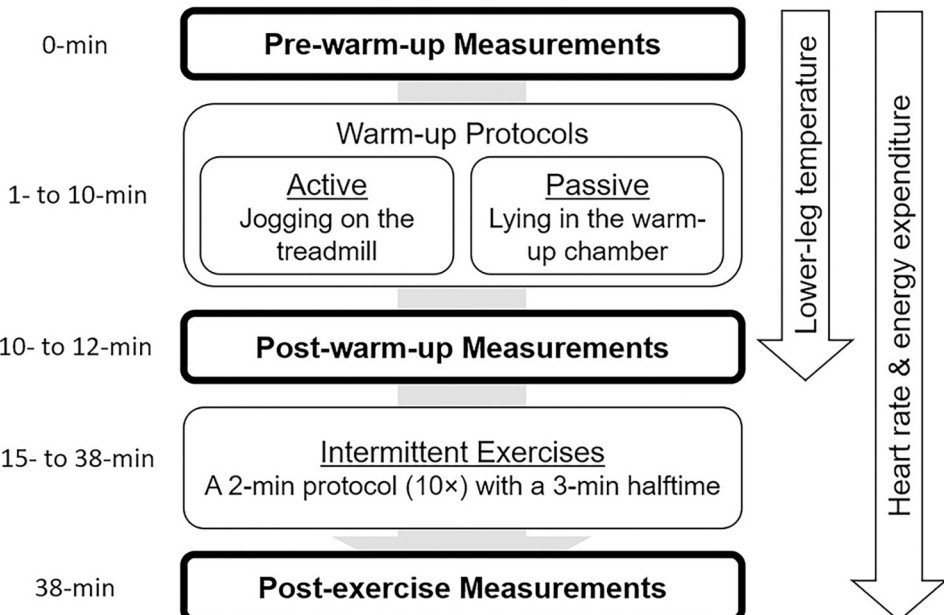

**Fig 1. Testing procedures.** The order of warm-up protocols was determined by coin tossing during the pre-warm-up measurements at the first session. Blood lactate concentration, subjective fatigue perception, and blood pressure were assessed during the measurements (indicated with bolded boxes). After the post-warm-up measurements, subjects moved to the track for the 3-min intermittent exercises.

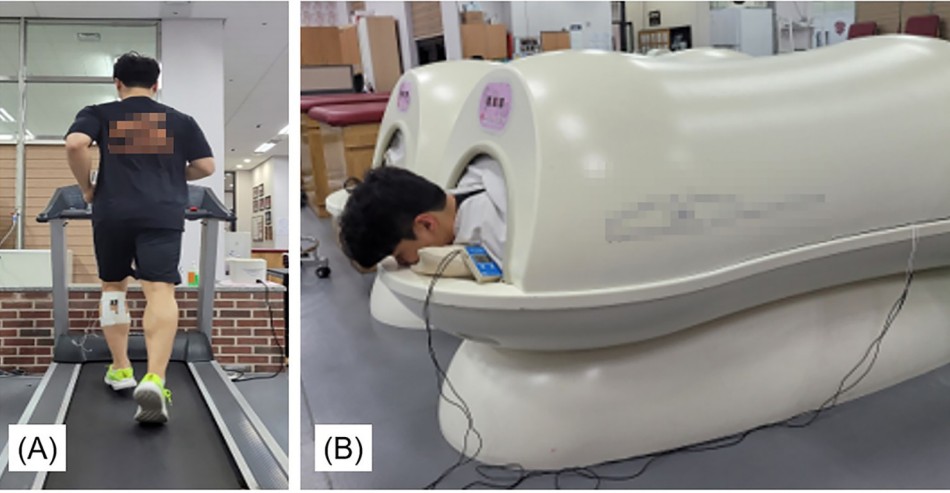

**Fig 2. 10-min warm-up protocols.** (A) Active warm-up: jogging on the treadmill (belt speed: 9.0 km/h at a 1% incline); (B) Passive warm-up: lying prone in the warm-up chamber (inner ambient temperature set as 35˚C).

volunteered to participate in this study. Subjects were excluded if they had any upper- or lower-extremity musculoskeletal pathology in the last six months, a history of lower-extremity surgery, or any general medical condition (e.g., diabetes, hypertension, etc.). Prior to study participation, subjects read the testing procedures and gave written informed consent, approved by the Kyung Hee University's Institutional Review Board (protocol#: KHGIRB-19-233; data collection period: October–November, 2019; data accessed for research purposes: November, 2019–March, 2020).

## Experimental procedures

Upon arrival at the laboratory, subjects read the testing procedures, gave informed consent, and changed into appropriate clothes (gym shorts and shirts). Afterward, subjects moved to the track for measurements of the baseline maximal vertical jump height.

Subjects were then asked to vertically jump off both legs as high as possible. Self-selected lower-extremity pre-stretch and upper-extremity single arm swinging were allowed when taking off. Practice trials were allowed until subjects were familiar with the movement and measurement. Two trials with a 30 sec rest interval were recorded and averaged for the 80% maximal vertical jumps for each subject. During this time, intermittent exercises were also introduced. The activities in this paragraph were omitted for the second visit.

Upon return to the laboratory, subjects sat on a chair for 10 min to achieve cardiovascular and thermal stability prior to the pre-warm-up measurements. During this time, the recording device for heart rate and lower-leg temperature were (for the muscle and skin temperature) attached to subjects. After assessing baseline heart rate and lower-leg temperature, other dependent measurements (blood lactate concentration and fatigue perception) were recorded.

Subjects then randomly performed one of the 10-min warm-up protocols at each session (Fig 1). For the active warm-up protocol, subjects jogged on the treadmill (constant belt speed: 9.0 km/h at a 1% incline: Fig 2A). For the passive warm-up protocol, subjects lay prone in a warm-up chamber. The chamber was preheated for 5 min prior to the warm-up protocol (inner ambient temperature was set to 35˚C: Fig 2B).

Post-warm-up measurements were taken (in the same manner as the pre-warm-up measurements) after the completion of each warm-up protocol. After detaching the thermistor probes on the lower-leg, subjects moved to the track for the intermittent exercises.

A total of 23 min of intermittent exercises, including a 3-min halftime, was performed. A previously established 2-min high-intensity intermittent exercise protocol [23] (intensity: % of maximal effort) consisting of 3 × 20 m walking (25%), 1 × vertical jumping (80%), 1 × 20 m sprinting (100%), 3 × 20 m jogging (60%), 1 × maximal vertical jumping (100%), and 3 × 20 m running (75%) was repeated ten times (a 3 min halftime was given after the fifth set). The intensity (% of maximal effort in walking, jogging, and running) was self-determined. The height of the 80% vertical jump was preset based on the maximal vertical jump height measured at the baseline measurement. Subjects performed a walking rest (self-paced) during the 3-min halftime. The subjects were asked to jog faster if they did not complete the 80% and 100% vertical jumps within 40 and 90 sec, from the beginning of each set, respectively. These cutoff times were based on our pilot study, which corresponded to the average completion pace of a 2-min intermittent exercise protocol. Walking rests were allowed if subjects completed a set of the intermittent exercise protocol earlier than 2 min. To examine subsequent athletic performance after two different warm-up protocols (active vs. passive), 20 m sprint times and the maximal vertical jump heights performed at each set of the intermittent exercises were recorded.

Post-exercise measurements were recorded in the same manner as the pre-warm-up measurements except for the lower-leg temperatures. Heart rate was continuously recorded from the pre-warm-up measurements through the post-exercise measurements. Heart rate data recorded throughout the experiment were used to estimate energy expenditure.

## Athletic performance

The jump-reach method was used to assess vertical jump heights. This jump test has been validated with the 3D motion analysis test [22] and shown to be reliable within (ICC>0.95) and between sessions (ICC = 0.86) as one of the exercise components when performing the intermittent exercises [23]. Subjects stood in the same foot position for maximal vertical jumps on a Vertec (Vertec, Sports Imports, Columbus, USA) and then raised (full scapular upward rotation with abduction) their dominant arm (the arm to throw a ball—all right-handed) directly overhead as high as possible to touch the plastic vanes while maintaining full extension of the lower-extremity and trunk.

A pair of timing sensors (Brower Timing System, Draper, USA) was used to record the 20 m sprint times. From the standing start position (1 m away from the first sensor), subjects were asked to sprint straight to the finish line where the second pair of sensors was located (20 m away from the first sensor). The maximal vertical jump heights were measured on Vertec as described in the baseline measurement.

## Lower leg temperature

To record the skin and muscle temperature of the lower-leg, two separate channels of thermistor probes (sampling rate: 60 Hz) connected to a digital logger thermometer (N543, NT logger, NKTC, Tokyo, Japan) were secured by a film dressing (Tegaderm Film, 3M, St. Paul, MN, USA). Each channel of the probes was attached to the distal 2/3 (skin) and proximal 1/3 (muscle) of the left medial gastrocnemius. To record muscle temperature (channel 2), the thermistor probe was covered by a wetsuit-material neoprene rubber (thickness: 3 mm, diameter: 2.5 cm) to minimise heat flow between skin and atmosphere. This measurement technique allowed researchers to record the temperature of the insulating area, which estimates muscle

temperature up to a depth of 2.2 cm [6, 24]. This method has been validated [6, 24] and has shown a high measurement reliability (ICC = 0.93) [25].

## Blood lactate concentration

To assess blood lactate concentration, a blood sample (0.7 μL) from the subjects' fingertips using a lancet needle (26-gauge, Lancets, Moa, Korea) was obtained on a test strip (Lactate Plus Lactate Test Strips, Nova Biomedical, Waltham, MA, USA). The blood sample strip was then inserted into a blood lactate concentration meter (Lactate Plus, Nova Biomedical, USA) to obtain the value of the blood lactate concentration. This method has been validated (r = 0.91) and has shown high measurement reliability (r = 0.99) [26].

## Fatigue perception, heart rate, energy expenditure

To assess fatigue perception, subjects were asked to mark their level of fatigue on a 10-cm visual analogue scale (VAS) [2, 14] labeled "unfatigued" at the left end and "fatigued" at the right end. To record heart rate, subjects wore a heart rate monitor (Polar H10 strap, Polar Electro Inc., NY, USA) on their chests, and the subjects' demographic information (sex, height, and mass) was entered on the mobile phone application (Polar Beat, Polar Electro Inc., New York, USA). After the heart rate monitor (sampled at 60 Hz) was wirelessly connected to the application, the heart rate value was recorded. This device has been validated and has shown high measurement reliability (ICC = 0.75–0.90) [27].

## Statistical analysis

The minimum required number of subjects were calculated based on the previously published data for 20 m sprint times [28] and maximal vertical jump heights [29]. When expecting an effect size (ES) of 0.78 (mean difference of 0.07 sec with a standard deviation of 0.09 sec) for 20 m sprint times and of 0.69 (mean difference of 4.3 cm with a SD of 6.3) for maximal vertical jump heights, the minimum sample size in each group was calculated as 13 and 17, respectively (an alpha of 0.05 and a beta of 0.2). To satisfy both variables, we recruited 18 subjects.

Mean and 95% confidence intervals were calculated from each dependent measurement value at each time point. To test immediate effects on warm-up protocols over time, a condition of time mixed-model analysis of variance (random variable: subject; fixed variables: condition and time) was performed: a $2 \times 38$ in heart rate and energy expenditure; a $2 \times 12$ in lower-leg temperature ($p \leq 0.001$); a $2 \times 3$ in blood lactate concentration and fatigue perception ($p < 0.05$); and a $2 \times 10$ in 20 m sprint times and maximal vertical jump heights ($p < 0.05$). Tukey-Kramer pairwise comparisons were performed as a post-hoc test. A statistical package SAS (Ver. 9.4, SAS Institute Inc., Cary, USA) was used for all tests. To determine practical significance, between-time effect sizes (ES = $[\bar{X}_1 - \bar{X}_2]$ / $\sigma_{\text{pooled}}$) were also calculated.

## Results

### Athletic performance

20 m sprint times between two warm-up protocols were not different over time (condition × time: $F_{9,323} = 1.26$, $p = 0.26$; condition effect: $F_{1,323} = 2.32$, $p = 0.13$: Fig 3A). Regardless of condition (time effect: $F_{9,323} = 4.15$, $p < 0.0001$: Fig 3B), subjects' sprint times at trial 1 (2.95 sec) were slower than those at trial 5 (2.87 sec, 3%, $p < 0.0001$, ES = 0.51), 6 (2.88 sec, 2%, $p = 0.001$, ES = 0.41), 8 (2.89 sec, 2%, $p = 0.01$, ES = 0.38), and 10 (2.88 sec, 2%, $p = 0.002$, ES = 0.42).

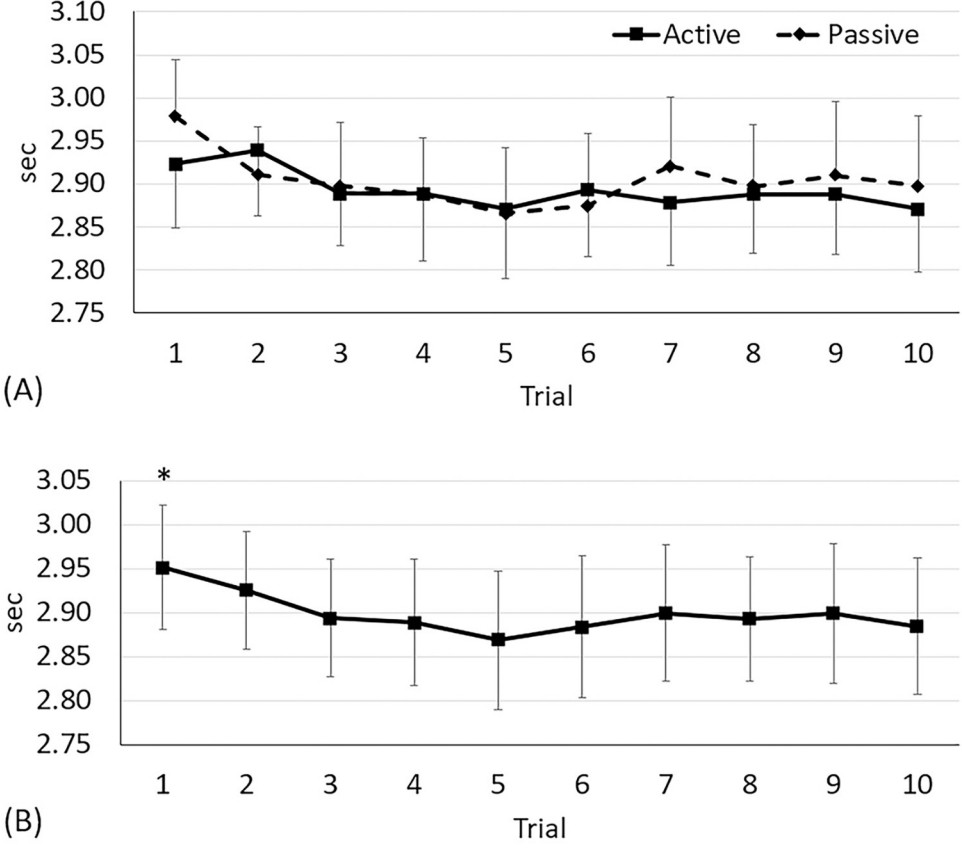

**Fig 3. Change in 20-m sprint time.** Values are mean and the upper and lower limits of 95% confidence intervals. (A) No condition × time interaction ($F_{9,323}$ = 1.26, $p$ = 0.26). (B) Time effect ($F_{9,323}$ = 4.15, $p<0.0001$). *Different from trials 5 ($p<0.0001$, ES = 0.51), 6 ($p$ = 0.001, ES = 0.41), 8 ($p$ = 0.01, ES = 0.38), and 10 ($p$ = 0.002, ES = 0.42).

Maximal vertical jump heights between warm-up protocols were different over time (condition × time: $F_{9,323}$ = 2.0, $p$ = 0.04). Subjects who performed the active warm-up protocol jumped higher by 4% than those who did the passive warm-up at trial 1 (active: 51.4; passive: 49.2 cm, $p$ = 0.04, ES = 0.31: Fig 4A), and there was no difference thereafter ($p<0.21$ for all tests). Regardless of time (condition effect: $F_{1,323}$ = 15.51, $p<0.0001$), subjects doing the active warm-up protocol (52.6 cm) jumped higher than the passive warm-up protocol (51.9 cm, $p<0.0001$, ES = 0.74). Regardless of condition (time effect: $F_{9,323}$ = 9.19, $p<0.0001$: Fig 4B), subjects' maximal vertical jump heights at trial 1 (50.3 cm) were increased at trial 3 (52.1 cm, 4%, $p$ = 0.001, ES = 0.26) through 10 (53.1 cm, 6%, $p<0.0001$, ES = 0.38). Maximal vertical jump heights at trial 2 (51.1 cm) were improved at trials 6 (53.2 cm, 4%, $p<0.0001$, ES = 0.30) through 10 (53.1 cm, 4%, $p$ = 0.0002, ES = 0.27).

## Lower-leg temperature

Lower-leg skin temperatures between the two warm-up protocols over time were different (condition × time: $F_{12,425}$ = 30.39, $p<0.0001$: Fig 5A). Subjects' performing the passive warm-up protocol showed higher lower-leg skin temperatures from the measurement at 1 (33.0°C) to 10 min (34.5°C), as compared with the active warm-up protocol (31.6 to 33.4°C; $p<0.0001$, ES = 1.12); two min after the completion of the warm-up protocols (12-min in Fig 5A), subjects who performed the active warm-up protocol (34.0°C) inversely showed 3% higher lower-

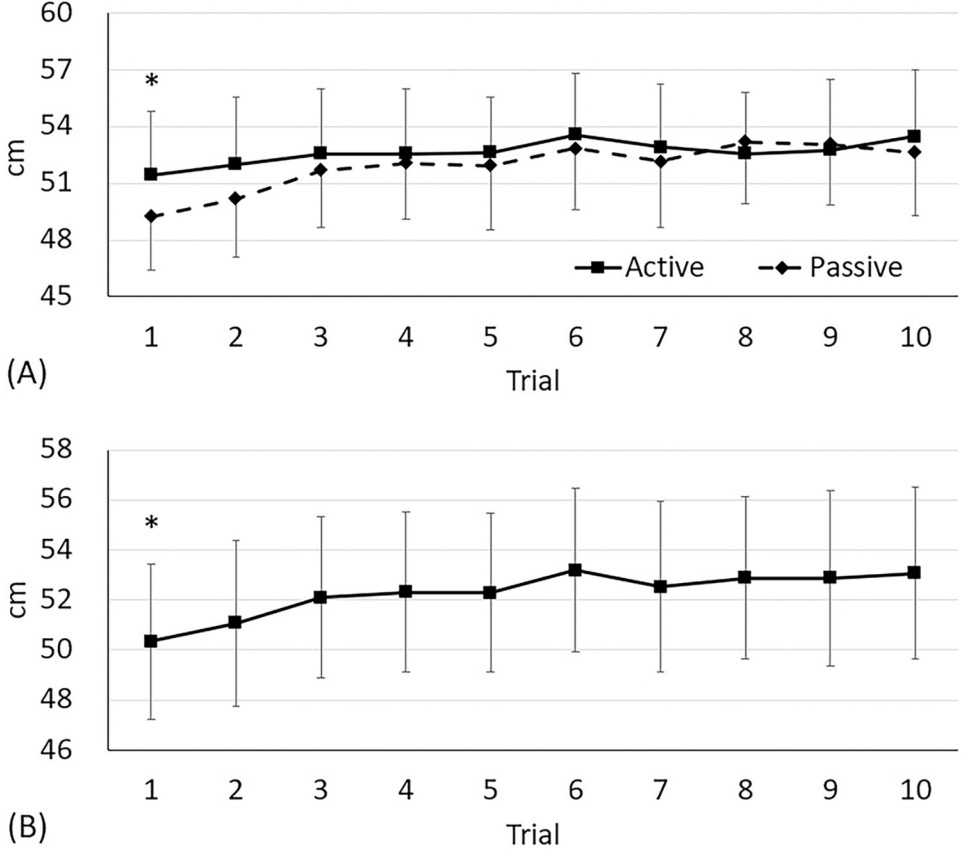

**Fig 4. Change in maximal vertical jump height.** Values are mean and the upper and lower limits of 95% confidence intervals. (A) Condition × time interaction ($F_{9,323} = 2.0$, $p = 0.04$). *Difference between the warm-up protocols at trial 1 ($p = 0.04$, ES = 0.33). (B) Time effect ($F_{9,323} = 9.19$, $p<0.0001$). *Different from trials 3 ($p = 0.001$, ES = 0.26) through 10 ($p<0.0001$, ES = 0.38).

leg skin temperature than those who did the passive warm-up protocol (33.0˚C, $p<0.0001$, ES = 1.62). Lower-leg muscle temperatures between two warm-up protocols over time were different (condition × time: $F_{12,425} = 13.99$, $p<0.0001$: Fig 5B). Subjects' performing the passive warm-up protocol showed higher lower-leg muscle temperatures from the measurement at 1 (33.5˚C) to 8 min (34.8˚C), as compared with the active warm-up protocol (32.7 to 34.2˚C; $p<0.0001$, ES = 1.36); two min after the completion of the warm-up protocols (12-min in Fig 5B), subjects who performed the voluntary warm-up protocol (35.0˚C) inversely showed 3% higher lower-leg muscle temperature than those who did the passive warm-up protocol (34.4˚C, $p<0.0001$, ES = 1.11).

## Blood lactate concentration

Blood lactate concentration between two warm-up protocols over time was different (condition × time: $F_{2,85} = 3.61$, $p = 0.03$: Table 1). Subjects who performed the active warm-up protocol (4.1 mmol/L) showed a higher blood lactate concentration by 173% than those who did the passive warm-up protocol (1.5 mmol/L) at the post-warm-up measurements ($p = 0.004$, ES = 1.69). However, there was no difference between the warm-up protocols post exercise (active: 10.2 mmol/L; passive: 9.4 mmol/L, $p = 0.87$, ES = 0.21).

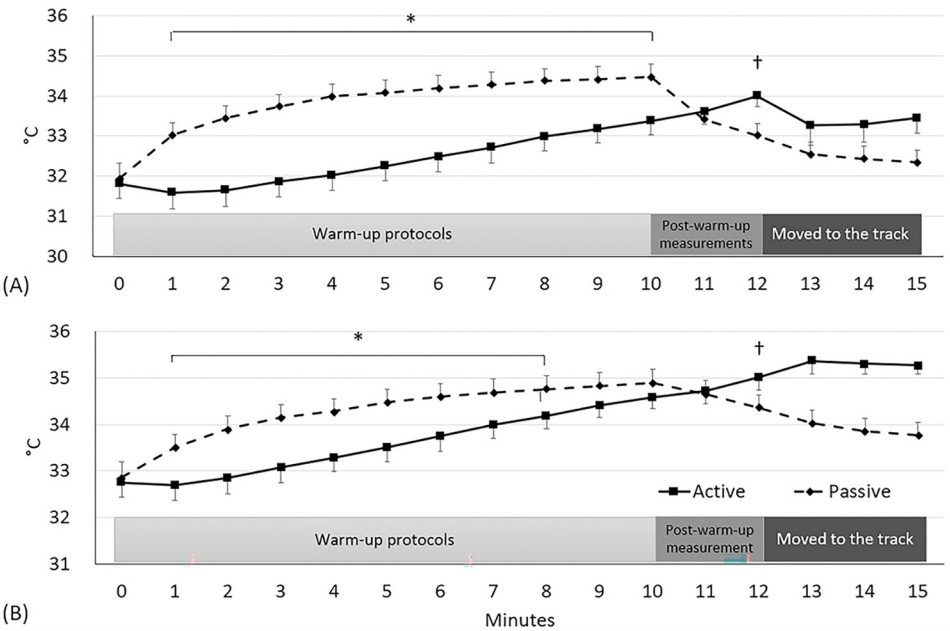

**Fig 5.** Change in the lower-leg temperature (A: skin; B: muscle). Values are mean and the upper and lower limits of 95% confidence intervals. Note that the data from 12 to 15 min (moving to the track) were separately collected in a pilot work (n = 8). (A) *Difference between the warm-up protocols within the specified time points (from 1 to 10 min: p<0.0001, an average ES = 2.12). †Difference between the warm-up protocols at post-warm-up (12 min: p<0.0001, an average ES = 1.62). (B) *Differences between the warm-up protocols within the specified time points (n = 18; from 1 to 8 min) (p<0.0001, an average ES = 1.47). †Difference between the warm-up protocols at post-warm-up (12 min: p<0.0001, an average ES = 1.11).

## Fatigue perception

Fatigue perception between the two warm-up protocols over time was not different (condition × time: $F_{2,85}$ = 0.5, $p$ = 0.61: Table 2). Regardless of time (condition effect: $F_{1,85}$ = 6.5, $p$ = 0.01), subjects with the passive warm-up protocol (4.7 cm) felt greater fatigue than the active warm-up protocol (4.0 cm, $p$<0.0001, ES = 0.71). Regardless of condition (time effect: $F_{2,85}$ = 16.68, $p$<0.0001), subjects' fatigue perception was increased at the post-warm-up (3.7 cm) and further increased at the post-exercise (5.6 cm, 51%, $p$<0.0001, ES = 3.86).

**Table 1. Changes in blood lactate concentration.**

| Unit: mmol/L | Active | Passive |
|:---:|:---:|:---:|
| Pre-warm-up | 1.4 (1.2 to 1.6) | 1.4 (1.1 to 1.7) |
| Post-warm-up | 4.1 (3.2 to 5.0)*† | 1.5 (1.1 to 1.9) |
| Post-exercise | 10.2 (8.6 to 11.8) | 9.4 (7.6 to 11.2)‡ |

Values are mean (the upper and lower limits of 95% confidence intervals).

\* Different from the passive warm-up protocol at post-warm-up ($p$ = 0.004, ES = 1.69).

† Difference from the active warm-up protocol at pre-warm-up ($p$ = 0.002, ES = 1.90) and post-exercise ($p$<0.0001, ES = 2.12).

‡ Different from the passive warm-up protocol at pre- ($p$<0.0001, ES = 2.80) and post-warm-up ($p$<0.0001, ES = 2.74).

**Table 2. Changes in subjective fatigue perception.**

| Unit: cm | Active | Passive |
|---|---|---|
| Pre-warm-up | 3.3 (2.4 to 4.2) | 4.3 (3.2 to 5.4) |
| Post-warm-up | 3.6 (2.7 to 4.5) | 3.9 (3.0 to 4.8) |
| Post-exercise | 5.1 (4.0 to 6.2)* | 6.0 (5.0 to 7.0)† |

Values are mean (the upper and lower limits of 95% confidence intervals).

* Difference from the active warm-up protocol at pre-warm-up ($p$ = 0.0009, ES = 0.83).

† Difference from the passive warm-up protocol at pre- ($p$ = 0.02, ES = 0.72) and post-warm-up ($p$ = 0.0009, ES = 1.00).

## Heart rate

Heart rate between the two warm-up protocols over time was different (condition × time: $F_{38,1309}$ = 123.72, $p$<0.0001). Subjects who performed the active warm-up protocol showed higher heart rates at the measurement times of 1 (137 bpm) and 15 min (118 bpm), as compared to those who performed the passive warm-up protocol (75 to 86 bpm; $p$<0.0001, ES = 4.94: Fig 6A).

## Energy expenditure

Energy expenditure between the two warm-up protocols over time was different (condition × time: $F_{38,1309}$ = 18.39, $p$<0.0001: Fig 6B). Subjects who performed the active

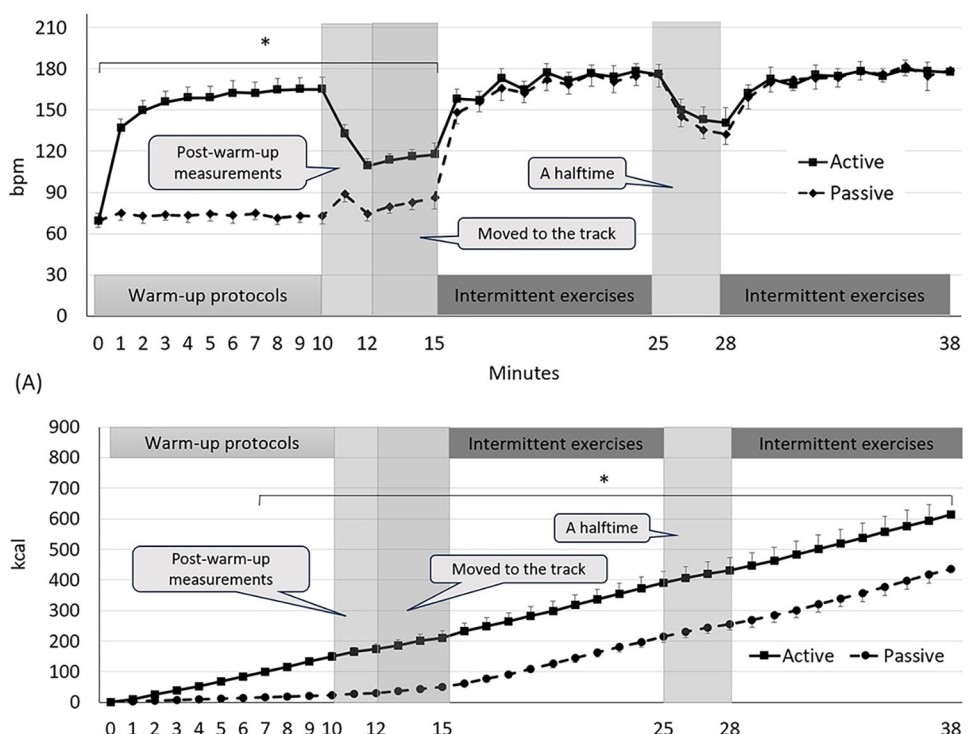

(A)

(B)

**Fig 6.** Change in heart rate (A) and energy expenditure (B). Values are mean and the upper and lower limits of 95% confidence intervals. (A) *Differences between the warm-up protocols within the specified time points (from 1 to 15 min: p<0.0001, an average ES = 4.94). (B) *Differences between the warm-up protocols within the specified time points (from 7 to 38 min: p<0.0001, an average ES = 3.69).

warm-up protocol showed higher energy expenditure at the measurement times of 7 (100 kcal) and 38 min (613 kcal), as compared with those who performed the passive warm-up protocol (17 to 436 kcal; $p < 0.0001$, ES = 3.69: Fig 6B).

## Discussion

The purpose of this study was to examine the effects of two different warm-up protocols with or without voluntary contraction on subsequent athletic performance during intermittent exercises. As we incorporated temperature-matched warm-ups, there was muscle temperature increase in the lower-leg. Our hypotheses were also accepted that subsequent athletic performance during the intermittent exercises did not differ between the active and passive warm-up protocols.

Although no significant interaction was found in athletic performance, a time effect (regardless of condition) on both the 20 m sprints (Fig 3B) and vertical jumps (Fig 4B) are practically meaningful in terms of warm-up duration. A warm-up activity should be a sufficient duration to increase body temperature and to prepare for peak performance while minimising energy depletion [4]. As compared with trial 1, subjects sprint performances improved at trial 5, and jump performances improved at trial 3; the best performance was recorded at trials 5 and 6 in the sprints (2.87 sec) and jumps (53.2 cm), respectively. The average heart rate for our active warm-up protocol was 150 bpm, which, converted, is 62% $VO_{2max}$ [30]. Therefore, the observed time effect in athletic performance supports the previous idea that the duration of 10 to 20 min would be optimal for a moderate intensity (60% to 80% $VO_{2max}$) warm-up [1]. Therefore, warm-up intensity is an important determinative factor. Previously, a high-intensity (80% $VO_{2max}$) warm-up resulted in a better jumping performance when compared with the moderate-intensity warm-up when the duration (15 min) was matched [2]. High-intensity warm-ups are often preferred for specific purposes such as the PAP effect [31], but a sufficient recovery time (e.g., 10-min) should be given prior to exercise to prevent pre-competition fatigue and muscle glycogen depletion [2].

Although thermal effects achieved between the active and passive warm-up did not differ, changing patterns in lower-leg temperature following each warm-up protocol were different (Fig 5). The lower-leg skin and muscle temperature in subjects performing the passive warm-up were higher at every minute during warm-up. However, 2 min after the completion of the warm-ups, the temperature (skin: 34.0°C; muscle: 35.0°C) in those performing the active warm-up was higher than the those performing the passive warm-up. This observation indicates a couple of important implications. First, the capacity for heat retention in muscle tissue is dependent on the method of heat gain. In our data, from the completion of the warm-up to just prior to the intermittent exercise (10 to 15 min in Fig 5A and 5B), lower-leg temperature in those performing the active warm-up (and gaining heat via metabolism) continuously increased, while the lower-leg temperature of those performing the passive warm-up (gaining heat via convection) rapidly dropped. To establish the heat reduction rate on each warm-up protocol, we performed a simple pilot study (n = 5) on how the lower-leg muscle temperature in a seated rest returns to the pre-warm-up level. As a result, it took 75 min and 41 min to return to baseline, which means that a reduction of 0.02°C and 0.04°C per minute occurred following the active and passive warm-ups, respectively. Summarising our data on temperature change, the active warm-up with voluntary muscle contraction had a low heating rate but a higher heat retention rate when compared with the passive warm-up. Second, therefore, this reverse pattern in temperature change should be considered when interpreting the subsequent athletic performance. To figure out the muscle temperature difference just prior to the intermittent exercise, we performed another pilot study (n = 8; Fig 5) where the temperature data

were continuously recorded while subjects were moving to the track (the place that they performed the intermittent exercises). The temperature difference was 1.5˚C (Fig 5B). In addition, the vertical jump height at the first trial was different (active: 51.4 cm; passive: 49.2 cm). Although the difference is negligeable (ES = 0.33), we believe that the vertical jump performance in each warm-up protocol was associated with the lower-leg temperature.

Blood lactate concentration is one of the commonly measured parameters to evaluate the level of physiological fatigue [32], which has shown a linear relationship to psychological fatigue (r = 0.75) as measured by a modified 10-cm visual analogue scale [33]. The levels of blood lactate concentration following each warm-up protocol (Table 1) should be influenced by the energy system used for each condition [32]. The magnitude of difference did not affect athletic performance, since the value of 4.1 mmol/L was insufficient to cause a significant muscle function reduction [34]. Unlike blood lactate concentration, interestingly, fatigue perception between the conditions at post-warm-up was not different. We believe that this could be from a perceptual stress caused by sudden physical demands for high-intensity exercises. Although not statistically significant, subjects who performed the passive warm-up showed greater perception of fatigue after the intermittent exercises (6.0 of 10 cm) than subjects who performed the active warm-up (5.1 of 10 cm). This result could be explained by the non-thermal effect from voluntary muscle contraction, since the degree of nerve impulse [1] and muscle stiffness (e.g., breaking the actin-myosin bonds) [1, 13] were different between warm-up protocols. For this reason, although it was insufficient to result in a different athletic performance, it was likely that subjects who performed the active warm-up were better prepared for the intermittent exercise. When the same workloads (intermittent exercises) were given, individuals performing the passive warm-up felt less perpetual preparation, resulting in a higher fatigue perception. This may suggest that avoiding a high level of relaxation and calmness prior to physically demanding activities. When considering there was no difference in athletic performance, however, coaches and athletes could optionally select either warm-up protocol.

Lastly, a few study limitations associated with methodological issues should be acknowledged. Our subjects were recreationally active (average exercise duration: 253 min per week) male collegiate students. We do not know whether we would see similar results from testing different populations, such as females or highly trained individuals. Note that the ambient temperature was 26˚C, which is slightly higher than a typical indoor athletic facility. This could have influenced athletic performance in our subjects. Cardiovascular and metabolic responses to exercise under a different weather condition would result in different thermal and non-thermal effects. Care should be taken when applying our warm-up protocols in the field. The measurement technique (insulation disk) used in our study is known to estimate inner tissue temperature up to 2.2 cm from the skin [6]. Our gastrocnemius muscle temperature (32.8˚C) is lower than those recorded using an invasive technique (thermistor insertion: 35.2˚C), which may suggest that the muscle temperature reported in our study could have been more superficial than what has been suggested. Some subjects performed vertical jumps before performing the passive warm-up. While we believe a 10 min seated rest was sufficient to washout, performing vertical jumps may have been a confounding factor in the intervention of passive warm-up. The jump-reach method for vertical jump heights in our study required body coordination skills (e.g., jump then touch the vanes). While the jump test might have been limited in assessing the pure jump ability (e.g., vertical displacement of the centre of mass), our study protocol using the intermittent exercises was not able to accommodate the methods known to be more accurate in measuring jump heights [22]. The running intensities of the intermittent exercise protocols were self-determined (e.g., 60% and 75% of maximal speed) with prescribed cutoff times (80% and 100% vertical jumps within 40 and 90sec, respectively, were completed). It is conceivable that there might have been variations in running

intensities when comparing to audio guidance [23]. However, all subjects were accustomed to the exercise protocols after a couple of sets and consistently finished with 5 to 10 sec left in each set. This indirectly indicates that running intensities of intermittent exercises within and between subjects were consistent.

## Conclusions

Subsequent sprint and jump performance during intermittent exercise in recreationally active young adult males after performing the active and passive warm-up protocols (duration- and muscle temperature-matched) were not different. According to our results, the non-thermal effect from the active warm-up appears to minimally affect athletic performance. Therefore, the passive warm-up could be suitable for sports that have breaks between competitions due to energy conservation [9, 33]. Athletes could also expect psychological benefits by proceeding with the performance routine [7] and performing pre-exercise mental training for the active and passive warm-ups, respectively [13]. With that in mind, strength and conditioning should select an appropriate warm-up method in consideration of the specific purpose, type of sports, and device availability.

## Supporting information

**S1 Data.**
(XLSX)

## Author Contributions

**Conceptualization:** Seunghee Lee, Junhyeong Lim, Jihong Park.

**Data curation:** Seunghee Lee, Junhyeong Lim.

**Formal analysis:** Seunghee Lee, Junhyeong Lim, Jihong Park.

**Methodology:** Jihong Park.

**Project administration:** Jihong Park.

**Supervision:** Jihong Park.

**Validation:** Jihong Park.

**Visualization:** Seunghee Lee, Junhyeong Lim, Jihong Park.

**Writing – original draft:** Seunghee Lee, Junhyeong Lim, Jihong Park.

**Writing – review & editing:** Seunghee Lee, Junhyeong Lim, Jihong Park.

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
