## [Decision Letter · Decision Letter 0]

4 Jul 2023

PONE-D-23-11192A warm-up strategy with or without voluntary contraction on athletic performance, lower-leg temperature, and blood lactate concentrationPLOS ONE

Dear Dr. Park,

Thank you for submitting your manuscript to PLOS ONE. After careful consideration, we feel that it has merit but does not fully meet PLOS ONE’s publication criteria as it currently stands. Therefore, we invite you to submit a revised version of the manuscript that addresses the points raised during the review process.

ACADEMIC EDITOR:Dear Authors, one expert in the field reviewed your manuscript, retrieving several majors you should consider during the revision process. Please submit your revised manuscript by Aug 18 2023 11:59PM. If you will need more time than this to complete your revisions, please reply to this message or contact the journal office at plosone@plos.org. Please include the following items when submitting your revised manuscript:A rebuttal letter that responds to each point raised by the academic editor and reviewer(s). You should upload this letter as a separate file labeled 'Response to Reviewers'.A marked-up copy of your manuscript that highlights changes made to the original version. You should upload this as a separate file labeled 'Revised Manuscript with Track Changes'.An unmarked version of your revised paper without tracked changes. You should upload this as a separate file labeled 'Manuscript'.

We look forward to receiving your revised manuscript.

Kind regards,

Emiliano Cè

Academic Editor

PLOS ONE

4. Please include captions for your Supporting Information tables at the end of your manuscript, and update any in-text citations to match accordingly. Please see our Supporting Information guidelines for more information: http://journals.plos.org/plosone/s/supporting-information.

Reviewers' comments:

Reviewer's Responses to Questions

**Comments to the Author**

1. Is the manuscript technically sound, and do the data support the conclusions?

Reviewer #1: No

2. Has the statistical analysis been performed appropriately and rigorously? 

Reviewer #1: Yes

3. Have the authors made all data underlying the findings in their manuscript fully available?

Reviewer #1: No

4. Is the manuscript presented in an intelligible fashion and written in standard English?

Reviewer #1: No

5. Review Comments to the Author

Reviewer #1: I would like to congratulate the authors for the effort and time spent conducting this study. This is an interesting study under the scope of the journal, and that could provide some practical applications for coaches. However, in my opinion, there are some major concerns regarding the methodology and evaluations that can compromise results. This is explained in specific comments and should be clarified before further revision.

Specific comments:

- The authors should clarify the use of some specific words. For example, what is the difference between voluntary and involuntary? Are the authors talking about active and passive warm-ups? Both exercises were voluntary. The main difference between them was performing physical exercise. This concept o voluntary or involuntary should be rethanked or clarified throughout the manuscript.

- English writing should be revised throughout the manuscript

- I think I understand the sentence, “Non-thermal effect from the warm-up activity may not be necessary for sprinting and jumping performance in recreationally active males.”. however, there are two negative sentences, and this should be rewritten.

- The hypothesis should be rewritten to replace the “similar” word. The hypothesis should be “different” or “not different”. Being not different is not the same as “similar”.

- The laboratory temperature is higher than usual for this kind of experiment (~26ºC). Could this have influenced the results? Please, clarify and, if needed, include it in the discussion.

- What were the level and/or experience of participants? Warm-up effects depend on the level and experience of participants, and this should be presented.

- Each participant performed a maximal vertical jump at baseline. This was performed before the voluntary and involuntary warm-up. So, there was physical exercise, together with the involuntary warm-up, in some participants (those randomly assigned to this condition on their first visit). Could this activation influence results? Please, discuss or clarify.

- Vertical jump was determined by vertec equipment, and this has strong limitations. Other equipment provides a more reliable measurement of the vertical jump. Moreover, in this kind of jump test, the participants' jump can be affected by several factors that should be considered, such as shoulder range of motion, arm swings, dominant and non-dominant hand, and bending. So, the authors should support their choice with a validation study of this procedure.

- Please, be specific to the time of collecting data (blood lactate and temperatures) after warm-up and after the main exercise to allow replication. And, if appropriate, what was used for analysis.

- The performance measure was recorded by using an intermittent exercise of 23 min. However, during this exercise, the intensity was self-determined, requiring a maximal intensity only in some parts of the test. This is kind of hard to understand if this should be a reliable measure of performance for several reasons. For example, how should we know that each one performed at 25% or 75% in both conditions, and some kind of saving energy exists and then contributes to the maximization of 100% intensity effort? Then, the authors report that to examine performance after two warm-up protocols, 20m sprint time and maximal vertical jump height at each set of intermittent exercises were recorded. So, as far as I understand, vertical jump and running at submaximal and maximal were performed and analysed. I am not sure about the validity of these measurements because of being submaximal and without any control (self-determined), and I am not sure about the influence of the effort management of each participant and whether each previous exercise will compromise any difference from previous warm-up conditions on subsequent exercise. This is a major issue of the study and should be clarified and discussed by the authors.

- Muscle temperature should be around 36/37º C at baseline, as presented in the literature. But the authors presented values below 33ºC. That is hypothermia, usually. All the range of values that are presented are lower than expected. There should be an explanation for these values, or please, consider removing them because it is a sign that some calibration was not correct.

6. PLOS authors have the option to publish the peer review history of their article (what does this mean?). If published, this will include your full peer review and any attached files.

Reviewer #1: No

---

## [Author Response · Author response to Decision Letter 0]

24 Jul 2023

Comments to the Author

1. Is the manuscript technically sound, and do the data support the conclusions?

Reviewer #1: No

2. Has the statistical analysis been performed appropriately and rigorously?

Reviewer #1: Yes

3. Have the authors made all data underlying the findings in their manuscript fully available?

Reviewer #1: No

4. Is the manuscript presented in an intelligible fashion and written in standard English?

Reviewer #1: No

5. Review Comments to the Author

Reviewer #1: I would like to congratulate the authors for the effort and time spent conducting this study. This is an interesting study under the scope of the journal, and that could provide some practical applications for coaches. However, in my opinion, there are some major concerns regarding the methodology and evaluations that can compromise results. This is explained in specific comments and should be clarified before further revision.

We would like to thank the editor and reviewer for critically reviewing our manuscript and providing valuable comments. Below, you can find a list of changes regarding your comments and our responses (highlighted in yellow). The changes in the revised manuscript were also highlighted in yellow. Thanks again for your valuable comments. 

Specific comments:

- The authors should clarify the use of some specific words. For example, what is the difference between voluntary and involuntary? Are the authors talking about active and passive warm-ups? Both exercises were voluntary. The main difference between them was performing physical exercise. This concept o voluntary or involuntary should be rethanked or clarified throughout the manuscript.

We agree with the reviewer that the terms voluntary and involuntary can be confused. We have defined active and passive warm-ups with appropriate references (Ln 10-14) and have renamed them throughout the manuscript. 

- English writing should be revised throughout the manuscript

The revised manuscript has been proofread.

- I think I understand the sentence, “Non-thermal effect from the warm-up activity may not be necessary for sprinting and jumping performance in recreationally active males.”. however, there are two negative sentences, and this should be rewritten.

This sentence has been revised (last sentence in the abstract).

- The hypothesis should be rewritten to replace the “similar” word. The hypothesis should be “different” or “not different”. Being not different is not the same as “similar”.

This has been revised, as the reviewer requested (Ln 52). 

- The laboratory temperature is higher than usual for this kind of experiment (~26ºC). Could this have influenced the results? Please, clarify and, if needed, include it in the discussion.

This has been added as a limitation as the reviewer requested (Ln 321-3).

- What were the level and/or experience of participants? Warm-up effects depend on the level and experience of participants, and this should be presented.

As the reviewer requested, the inclusion criterion on the activity level in our subjects was added in the subject description paragraph (Ln 74-6). The average exercise duration was also added in the limitation paragraph (Ln 318-21).

- Each participant performed a maximal vertical jump at baseline. This was performed before the voluntary and involuntary warm-up. So, there was physical exercise, together with the involuntary warm-up, in some participants (those randomly assigned to this condition on their first visit). Could this activation influence results? Please, discuss or clarify.

This has been added as a limitation, as the reviewer requested (Ln 330-2).

- Vertical jump was determined by vertec equipment, and this has strong limitations. Other equipment provides a more reliable measurement of the vertical jump. Moreover, in this kind of jump test, the participants' jump can be affected by several factors that should be considered, such as shoulder range of motion, arm swings, dominant and non-dominant hand, and bending. So, the authors should support their choice with a validation study of this procedure.

We have added a statement regarding the validation and reliability of the jump-reach test in the similar subject population (recreationally active college students: Ln 86-90). Our subjects were all right-dominant, which has been updated in the methods section (Ln 92). Although it is a valid and reliable test tool, we agree with the reviewer’s point here that the jump-reach method requires body coordination. We have added this as a potential limitation (332-7). Our laboratory has previously used this device to collect vertical jump data on both athletic and general populations and has published several papers using said data (papers that are not cited here due to blind review): for three of those recently published papers, high reliabilities (ICC of 0.95-0.98; 0.94-0.97; 0.94-0.98) were reported. Although the current study design did not allow us to calculate ICC values (no baseline values were collected), we believe that the measurement consistency issue was minimal. 

- Please, be specific to the time of collecting data (blood lactate and temperatures) after warm-up and after the main exercise to allow replication. And, if appropriate, what was used for analysis.

To clarify the measurement time points, we have modified Figure 1 and the related captions. The analysis processes for blood lactate levels and muscle temperature were described in the method section (Ln 109-126). 

- The performance measure was recorded by using an intermittent exercise of 23 min. However, during this exercise, the intensity was self-determined, requiring a maximal intensity only in some parts of the test. This is kind of hard to understand if this should be a reliable measure of performance for several reasons. For example, how should we know that each one performed at 25% or 75% in both conditions, and some kind of saving energy exists and then contributes to the maximization of 100% intensity effort? Then, the authors report that to examine performance after two warm-up protocols, 20m sprint time and maximal vertical jump height at each set of intermittent exercises were recorded. So, as far as I understand, vertical jump and running at submaximal and maximal were performed and analysed. I am not sure about the validity of these measurements because of being submaximal and without any control (self-determined), and I am not sure about the influence of the effort management of each participant and whether each previous exercise will compromise any difference from previous warm-up conditions on subsequent exercise. This is a major issue of the study and should be clarified and discussed by the authors.

The reviewer has a valid point here. Originally (Loughborough Intermittent Shuttle Test, Nicholas, 2000), the intensities of various jumping and running movements were controlled by % VO2max, which was guided by audio feedback. This method has been updated to include the 80% maximal vertical jumps (Welsh, 2002) and was modified again by adding the 80% and 100% maximal vertical jumps (Kim, 2016). From our pilot work, we found that the use of self-determined speeds alongside prescribed cutoff times (Ln 146-9: the 80% and 100% maximal vertical jumps were completed within 40 and 90 sec, respectively) assisted subjects to maintain the running intensity during the intermittent exercises. For example, all subjects were accustomed to the running intensities after a couple of sets and had 5 to 10 sec left between sets. Therefore, we believe the intensities of intermittent exercises within and between subjects were consistent. This has been added in the limitation section (Ln 338-45).

- Nicholas et al. The Loughborough Intermittent Shuttle Test: A field test that simulates the activity pattern of soccer. Journal of Sports Science. 2000; 187: 97-104.

- Welsh et al. Carbohydrates and physical/mental performance during intermittent exercise to fatigue. Medicine & Science in Sports & Exercise. 2002; 34: 723-31.

- Kim et al. Joint cooling does not hinder athletic performance during high-intensity intermittent exercise. International Journal of Sports Medicine. 2016; 37: 641-46.

- Muscle temperature should be around 36/37º C at baseline, as presented in the literature. But the authors presented values below 33ºC. That is hypothermia, usually. All the range of values that are presented are lower than expected. There should be an explanation for these values, or please, consider removing them because it is a sign that some calibration was not correct.

We recorded the medial gastrocnemius for the muscle temperature. We believe that a body temperature around 36/37ºC is referred to as the core temperature, but this may not be the temperature in the distal muscle. Previously reported gastrocnemius muscle temperatures are listed below. 

Author 

(year) Title Muscle temperature

(body part, technique, depth of measurement)

Bang 

(2022) A 7-min halftime jog mitigated the reduction in sprint performance for the initial 15-min of the second half in a simulated football match 31.1 ± 2.4°C

(gastrocnemius, noninvasive, 2.2 cm)

Lim 

(2021) Comparison of 4 different cooldown strategies on lower-leg temperature, blood lactate concentration, and fatigue perception after intense running 33.1 ± 0.7°C

(gastrocnemius, noninvasive, 2.2 cm)

Muraoka

(2008) Effects of muscle cooling on the stiffness of the human gastrocnemius muscle in vivo 34.0 ± 0.8°C

(gastrocnemius, noninvasive, 2.2 cm)

Ichinoseki-Sekine

(2007) Changes in muscle temperature induced by 434 MHz microwave hyperthermia 35.0 ± 1.3°C

(vastus lateralis, noninvasive, 2 cm)

Garrett

(2000) Heat distribution in the lower leg from pulsed short-wave diathermy and ultrasound treatments 35.2 ± 1.0°C

(gastrocnemius, invasive, 3.0 cm)

Draper

(1995) Rate of temperature decay in human muscle following 3 MHz ultrasound: The stretching window revealed 33.8 ± 1.3°C

(gastrocnemius, invasive, 1.2 cm)

Our data (32.8 ± 0.7°C) are within the ranges of previously reported data that were collected using noninvasive techniques (Bang, 2022; Lim, 2021; Muraoka, 2008). While the data collected using invasive techniques (Garrett, 2000; Draper, 1995) are considered more accurate, it seems to be that the depth of measurement is the main factor in determining the temperature. There could be a possibility of underestimation when using noninvasive methods since these noninvasive techniques are known to estimate the inner tissue temperature at a depth of 2.2 cm. We have discussed this issue as a limitation (Ln 326-31).

6. PLOS authors have the option to publish the peer review history of their article (what does this mean?). If published, this will include your full peer review and any attached files.

Do you want your identity to be public for this peer review? For information about this choice, including consent withdrawal, please see our Privacy Policy.

Reviewer #1: No

---

## [Decision Letter · Decision Letter 1]

10 Oct 2023

PONE-D-23-11192R1A warm-up strategy with or without voluntary contraction on athletic performance, lower-leg temperature, and blood lactate concentrationPLOS ONE

Dear Dr. Park,

Thank you for submitting your manuscript to PLOS ONE. After careful consideration, we feel that it has merit but does not fully meet PLOS ONE’s publication criteria as it currently stands. Therefore, we invite you to submit a revised version of the manuscript that addresses the points raised during the review process.

ACADEMIC EDITOR:Dear Authors, your manuscript was newly revised by one expert in the field, who still found some minor points you should consider in the revision process.

We look forward to receiving your revised manuscript.

Kind regards,

Emiliano Cè

Academic Editor

PLOS ONE

Journal Requirements:

Reviewers' comments:

Reviewer's Responses to Questions

**Comments to the Author**

1. If the authors have adequately addressed your comments raised in a previous round of review and you feel that this manuscript is now acceptable for publication, you may indicate that here to bypass the “Comments to the Author” section, enter your conflict of interest statement in the “Confidential to Editor” section, and submit your "Accept" recommendation.

Reviewer #1: All comments have been addressed

2. Is the manuscript technically sound, and do the data support the conclusions?

Reviewer #1: Yes

3. Has the statistical analysis been performed appropriately and rigorously? 

Reviewer #1: Yes

4. Have the authors made all data underlying the findings in their manuscript fully available?

Reviewer #1: No

5. Is the manuscript presented in an intelligible fashion and written in standard English?

Reviewer #1: Yes

6. Review Comments to the Author

Reviewer #1: Congratulations to the authors, for their efforts in clarifying some important issues and, in my opinion, improving the quality of the manuscript. I still think that the manuscript has some important gaps that are related to the study design and procedures, as indicated in my previous review. However, the authors provided some important limitations that should be considered by the reader, when analyzing this manuscript.

However, some minor issues should be addressed, specifically:

- In the abstract, the authors continue to report that “voluntary contraction appeared to produce similar effects…” and the non-existence of difference is not the same as similar. Please, revise throughout the manuscript (i.e., discussion and conclusion).

- The background is clear and the introduction is supported by literature, but some evidence can be included to support it. For example, the authors reported that “Specifically, a 2% (0.7°C) core temperature increase improved sprinting and jumping performance by 6% and 4%, respectively [4]; an increase in working muscle temperature of just 1°C could affect the variation of athletic performance up to 5%. In general, muscle temperature increases rapidly during the first 3 to 5 min of warm-up and reaches a plateau after 10 to 20 min of activity [6]”. This same increase in core temperature happens in other sports, such as swimming after warm-up (e-.g., doi: 10.1519/JSC.0000000000001701), but the authors should also emphasize that this temperature can reduce immediately after ending warm-up (I.e., doi: DOI: 10.1111/j.1600-0838.2004.00349.x; doi: 10.1519/JSC.0000000000001701), and this should be discussed.

- I suggest that the authors divide the experimental procedures by each variable that was assessed so that can be easily understood.

- Please, remove data results from the discussion. New results should be included in results sessions or not discussed, and the results presented in the previous section should not be duplicated.

7. PLOS authors have the option to publish the peer review history of their article (what does this mean?). If published, this will include your full peer review and any attached files.

Reviewer #1: No

---

## [Author Response · Author response to Decision Letter 1]

2 Nov 2023

Dear editor and reviewers:

We would like to thank you for taking time to review our manuscript and provide valuable comments. We believe that revisions made in response to your comments have clarified and strengthened the manuscript. Below is a list of changes regarding the requested revisions of the manuscript (our responses are highlighted in yellow). Changes and revisions are also highlighted in yellow in the manuscript. 

Thank you again for reviewing our manuscript. We look forward to hearing from you.

Sincerely,

Authors

 

Dear Dr. Park,

Thank you for submitting your manuscript to PLOS ONE. After careful consideration, we feel that it has merit but does not fully meet PLOS ONE’s publication criteria as it currently stands. Therefore, we invite you to submit a revised version of the manuscript that addresses the points raised during the review process.

ACADEMIC EDITOR:

Dear Authors, 

your manuscript was newly revised by one expert in the field, who still found some minor points you should consider in the revision process.

We look forward to receiving your revised manuscript.

Kind regards,

Emiliano Cè

Academic Editor

PLOS ONE

Journal Requirements:

We have carefully checked but not found any retracted articles in the reference list. As requested, we have revised the reference list (Please see yellow-highlighted); 

removed the ref. #36 (Ln 352) and replaced by the ref. #7.

Reviewers' comments:

Reviewer's Responses to Questions

Comments to the Author

1. If the authors have adequately addressed your comments raised in a previous round of review and you feel that this manuscript is now acceptable for publication, you may indicate that here to bypass the “Comments to the Author” section, enter your conflict of interest statement in the “Confidential to Editor” section, and submit your "Accept" recommendation.

Reviewer #1: All comments have been addressed

2. Is the manuscript technically sound, and do the data support the conclusions?

Reviewer #1: Yes

3. Has the statistical analysis been performed appropriately and rigorously?

Reviewer #1: Yes

4. Have the authors made all data underlying the findings in their manuscript fully available?

Reviewer #1: No

5. Is the manuscript presented in an intelligible fashion and written in standard English?

Reviewer #1: Yes

6. Review Comments to the Author

Reviewer #1: Congratulations to the authors, for their efforts in clarifying some important issues and, in my opinion, improving the quality of the manuscript. I still think that the manuscript has some important gaps that are related to the study design and procedures, as indicated in my previous review. However, the authors provided some important limitations that should be considered by the reader, when analyzing this manuscript.

However, some minor issues should be addressed, specifically:

- In the abstract, the authors continue to report that “voluntary contraction appeared to produce similar effects…” and the non-existence of difference is not the same as similar. Please, revise throughout the manuscript (i.e., discussion and conclusion).

As the reviewer requested, we have revised the related sentences in the abstract, discussion (Ln 250-251; 268-269; 302), and conclusion (Ln 346-348). We have also made additional revisions to the conclusion (please see highlighted). 

- The background is clear and the introduction is supported by literature, but some evidence can be included to support it. For example, the authors reported that “Specifically, a 2% (0.7°C) core temperature increase improved sprinting and jumping performance by 6% and 4%, respectively [4]; an increase in working muscle temperature of just 1°C could affect the variation of athletic performance up to 5%. In general, muscle temperature increases rapidly during the first 3 to 5 min of warm-up and reaches a plateau after 10 to 20 min of activity [6]”. This same increase in core temperature happens in other sports, such as swimming after warm-up (e-.g., doi: 10.1519/JSC.0000000000001701), but the authors should also emphasize that this temperature can reduce immediately after ending warm-up (I.e., doi: DOI: 10.1111/j.1600-0838.2004.00349.x; doi: 10.1519/JSC.0000000000001701), and this should be discussed.

While appreciate the reviewer’s suggestion with the recommended articles here, we would like to keep the current introduction. The first paragraph (Ln 2-13) introduces the magnitudes of body temperature increase after warm-up and defines the active and passive warm-up. Since the second paragraph (Ln 14-28) talks about thermal and non-thermal effects, we thought that adding “an immediate temperature reduction after warming-up” is a different topic. Our decision was to increase readability and to link with the second paragraph. If the reviewer still thinks that “an immediate temperature reduction after warm-up” needs to be mentioned, we will be happy to add.

- I suggest that the authors divide the experimental procedures by each variable that was assessed so that can be easily understood.

As the reviewer requested, we have added appropriate subheadings. 

- Please, remove data results from the discussion. New results should be included in results sessions or not discussed, and the results presented in the previous section should not be duplicated.

We have removed most data results in the discussion section. We left cited Table and Figures to provide references. We have decided to leave some of the data results for better understanding (please see highlighted). If the reviewer thinks that they still need to be removed, we will be happy to do so.

7. PLOS authors have the option to publish the peer review history of their article (what does this mean?). If published, this will include your full peer review and any attached files.

Do you want your identity to be public for this peer review? For information about this choice, including consent withdrawal, please see our Privacy Policy.

Reviewer #1: No

---

## [Decision Letter · Decision Letter 2]

27 Nov 2023

A warm-up strategy with or without voluntary contraction on athletic performance, lower-leg temperature, and blood lactate concentration

PONE-D-23-11192R2

Dear Dr. Park,

We’re pleased to inform you that your manuscript has been judged scientifically suitable for publication and will be formally accepted for publication once it meets all outstanding technical requirements.

Kind regards,

Emiliano Cè

Academic Editor

PLOS ONE

Additional Editor Comments (optional):

Reviewers' comments:

Reviewer's Responses to Questions

**Comments to the Author**

1. If the authors have adequately addressed your comments raised in a previous round of review and you feel that this manuscript is now acceptable for publication, you may indicate that here to bypass the “Comments to the Author” section, enter your conflict of interest statement in the “Confidential to Editor” section, and submit your "Accept" recommendation.

Reviewer #1: (No Response)

2. Is the manuscript technically sound, and do the data support the conclusions?

Reviewer #1: Yes

3. Has the statistical analysis been performed appropriately and rigorously? 

Reviewer #1: Yes

4. Have the authors made all data underlying the findings in their manuscript fully available?

Reviewer #1: Yes

5. Is the manuscript presented in an intelligible fashion and written in standard English?

Reviewer #1: Yes

6. Review Comments to the Author

Reviewer #1: The authors did not address all my suggestions and recommendations, but justified the options. This is not a problem for me, as the suggestions were only to improve the quality of the manuscript.

7. PLOS authors have the option to publish the peer review history of their article (what does this mean?). If published, this will include your full peer review and any attached files.

Reviewer #1: No

---

## [Editor Report · Acceptance letter]

3 Jan 2024

PONE-D-23-11192R2 

PLOS ONE

Dear Dr. Park, 

I'm pleased to inform you that your manuscript has been deemed suitable for publication in PLOS ONE. Congratulations! Your manuscript is now being handed over to our production team.

Kind regards, 

on behalf of

Prof. Emiliano Cè 

Academic Editor

PLOS ONE